# Scalable and sustainable synthesis of chiral amines by biocatalysis
Matthew J. Takle [1,4], David M. Maurer [2], Philipp Staehle[2], Joachim Dickhaut[2], Christian Holtze[2], Klaus Hellgardt[3] & King Kuok Mimi Hii [1] ✉

In recent years, industrial biocatalysis has significantly advanced, largely due to innovations in DNA sequencing, bioinformatics, and protein engineering. However, the challenge of implementing biocatalysis at an industrial scale while ensuring sustainability and cost-effectiveness remains a critical barrier. This study presents the development of a flash thermal racemization protocol for chemoenzymatic dynamic kinetic resolution (FTR-CE-DKR) of chiral amines, encompassing an investigation of substrate scope, catalyst screening, and optimization studies. The outcomes of this research facilitated the successful scale-up of an industrially relevant amide within a recycle-batch platform, achieving unprecedented scales of up to 100 grams and space-time yield (STY) values of up to 73.2 g L$^{-1}$ h$^{-1}$. Furthermore, the process exhibited very favorable sustainability metrics when benchmarked against previous reports, including atom economy, reaction mass efficiency, and process mass intensity. These findings represent a significant milestone in the biocatalytic production of optically active amines.

Chiral amines are important for the manufacturing of high-value chemicals, including active pharmaceutical ingredients (APIs) and agrochemicals. Consequently, synthesis of these chiral building blocks has always been a highly topical and industrially relevant subject, with transition metal-catalyzed asymmetric (transfer) hydrogenation[1,2] and biocatalysis[3,4] as two of the most common approaches. The former may be considered as a more mature industrial technology[5–8], where TON > 10$^7$ and TOF of >200 s$^{-1}$ were reported recently[9]. In comparison, biocatalytic reactions are less productive and are more challenging to scale up[10–13]. Nevertheless, enzymes are generally regarded to be more sustainable than transition metal catalysts[14], as it avoid the use of precious metal catalysts and expensive chiral ligands.

To date, one of the most successful industrial biocatalytic processes is the ChiPros process, developed by BASF in 2001[15]. For the past two decades, the technology has been adopted across several manufacturing sites for the production of optically active amines, including a herbicide intermediate, with a total annual production of between 3 and 5 kt[16]. The overall process comprises of 3 steps (demonstrated in Scheme 1 using a generic 1-arylethylamine, **1**):

1) The biocatalytic kinetic resolution of a racemic mixture of amine (*RS*)-**1** in the presence of a lipase and a methoxyacetic acid ester, **3**, as the acylating agent. The reaction proceeds to the theoretically maximum 50%

conversion, affording the resolved amide (*R*)-**2** in near-perfect optical purity of 99.5% ee. The mixture of (*R*)-**2** and (*S*)-**1** can be separated by distillation.

2) (*R*)-**2** is hydrolyzed under basic conditions to liberate the amine (*R*)-**1**, which can be purified by distillation, while the sodium salt of methoxyacetic acid byproduct can be recovered and recycled.

3) Up to this point of the process, the racemate has been kinetically resolved into its constituent enantiomers. In practice, however, one enantiomer is often more valuable than the other. To minimize waste, the unwanted enantiomer can be racemized in the presence of catalytic amounts of acetophenone and a strong base via a Schiff base anion. After distillation, the resultant racemate can be combined with the original feedstock in step 1 for the lipase-catalyzed kinetic resolution.

While the ChiPros process is highly efficient, the need for a separate process to recycle the unwanted enantiomer (step 3) requires additional units of operation and resources, including multiple energy-intensive distillations for the isolation and recovery of amines. Furthermore, the method for recycling the unwanted enantiomer is not universally applicable, especially for secondary or tertiary amines.

In contrast, a chemoenzymatic dynamic kinetic resolution (CE-DKR) process can potentially offer a more sustainable alternative, whereby the lipase-catalyzed kinetic resolution is combined with a (chemo)catalyst that can selectively racemize the chiral center of the amine, such that the desired

[1]Department of Chemistry, Molecular Sciences Research Hub, Imperial College London, London, UK. [2]BASF SE, Ludwigshafen, Germany. [3]Department of Chemical Engineering, Imperial College London, London, UK. [4]Present address: School of Chemistry, University of Leeds, Leeds, UK. ✉ e-mail: mimi.hii@imperial.ac.uk

**Scheme 1 |** Overview of BASF's ChiPros process. Synthesis of chiral primary amines via sequential kinetic resolution, hydrolysis, and racemization.

### 1. lipase-catalyzed kinetic resolution

### 2. Recovery of *R*-enantiomer

### 3. Racemization of (unwanted) *S*-enantiomer

**Scheme 2 |** Dynamic kinetic resolution of chiral amines. Combining enzymatic resolution with racemization to enable complete conversion into a single enantiomer.

**Scheme 3 |** Mechanistic pathway for transition metal-catalyzed racemization of chiral amines. Reversible redox steps and competing side reactions.

enantiomer can be obtained as the resolved amide with a theoretical yield of 100% (Scheme 2).

In principle, chiral amines containing an α-H can be racemized by reversible hydrogenation-dehydrogenation steps (Scheme 3). A number of homogenous and heterogeneous[17,18] transition metal catalysts, including Ni[19,20], Pd[21–25], Ru[26,27], Ir[28–31], and even certain enzymes[32–35], have been reported to be effective in scrambling the stereocenter. For the racemization

of primary amines, the imine intermediate **I** could be arrested by another molecule of the primary amine to form secondary imine **4** and amine **5**; the latter as a mixture of diastereoisomers. To suppress the formation of these side products, racemization is often conducted under an atmosphere of H₂, or in the presence of a base additive. However, the presence of these additives shifts the position of the equilibrium, resulting in slower rate of racemization and prolonged reaction times.

**Table 1 | Previously reports of CE-DKR of chiral amine 1a using compartmentalized reactors in continuous flow**

| Entry | Product | Reactor 1 | Reactor 2 | Additive | Scale, conditions | STY (mmol $L^{-1}$ $h^{-1}$)[a] | Ref. |
|---|---|---|---|---|---|---|---|
| 1 | | Novozym 435 (PBR), RT | Pd/BaSO$_4$ (CSTR), 70 °C | Ammonium formate, Na$_2$CO$_3$ | 3.2 mmol, 10 h, recycle batch | 20.2 | 46 |
| 2 | | CALB-TDP10 (PBR), 60 °C | CALB-TDP10, Pd/AMP-KG (mixed-bed PBR), 60 °C | Ammonium formate | 138 mM, 5 µL min$^{-1}$, single pass | 4.14 | 47 |
| 3 | | Novozym 435 (PBR), 40 °C | Pd/γ-Al$_2$O$_3$ (PBR), 140 °C | None | 8.25 mmol, 1 h, recycle batch | 7.35 | 48 |

[a]Calculated using Eq.1.

Given that lipase-catalyzed resolution of amines is very efficient with selectivity factors, E, often >200, much of the research has been focused on the design of racemization catalysts that can operate synergistically under the same conditions as the biocatalyst in 'one-pot'[36–44]. To avoid thermal degradation of the biocatalyst, the one-pot reactions were invariably performed at temperatures that are sub-optimal for the metal-catalyzed racemization, resulting in slow processes that are not useful for scale up. In a seminal review by Verho and Bäckvall in 2015[45], the compatibility between the enzyme and the racemization catalyst was identified as a critical issue for CE-DKR processes.

Prior to our work, there were only two attempts to compartmentalize the catalysts, so that each of them can operate independently under optimized and kinetically compatible conditions (Table 1). The first example of this approach was reported by De Miranda et al.[46], where kinetic resolution of 1-phenylethylamine (**1a**) was achieved by a lipase in a packed bed reactor (PBR) at room temperature, while the racemization was performed in a continuous-stirred tanked reactor (CSTR) using a supported Pd catalyst at 70 °C (Entry 1, Table 1). In this hybrid system, the addition of ammonium formate was necessary to suppress the formation of side-products, but catalyst deactivation was a significant issue. Following this, Farkas et al. reported a single-pass flow system[47], where a mixture of the racemic **1a**, acylating reagent and ammonium formate was first passed through a PBR of the sol-gel immobilized lipase, followed by a mixed-bed PBR containing both the lipase and the racemization catalysts (Entry 2). However, as both reactors were operated isothermally at 60 °C, it does not offer any significant advantages over a "one-pot" system, and the productivity remained low. More recently, we disclosed an original method of achieving CE-DKR in a flow system, where the racemization of amine was facilitated under "flash thermal racemization" (FTR) conditions without any extraneous additives (Entry 3)[48]. This represented an original approach, whereby good reactivity and selectivity of amine racemization can be achieved by applying a short residence time at high temperature (9 s, 140 °C). Using this method, we were able to perform the CE-DKR of amines on a 1 g scale in 1 h.

In order to provide a fair comparison between (recycle)batch and single-pass operations, space-time-yield calculations were performed using the following equation (Eq. 1), where the total volume of solvent used to produce a specific quantity of product is utilized, rather than the volume of the reactor(s). Given that the use of solvent is always a substantial contributor to process mass intensity of a process (PMI), the resultant STY value will also reflect the sustainability of the process. Using this comparison, the FTR methodology does not compare so favorably, due to the large amount of solvent used, although the amount of Pd used is much lower than the other two processes.

$$\mathrm{STY}\left(\mathrm{mmol\,L^{-1}h^{-1}}\right) = \frac{\text{amount of kinetically resolved amide (mmol)}}{\text{total volume of solvent (L)} \times \text{time (h)}}$$

(1)

Subsequently, the FTR process was explored further using DoE and transient flow methods, where relationships between temperature, amine concentration and flow rate with e.e. (%) and selectivity (%) were delineated (3 factors, 2 responses)[49]. The study revealed that the rate of racemization is strongly dependent on temperature, while the selectivity of the process was dependent on both temperature and flow rate/residence time. Remarkably, the initial concentration of the amine was not found to have a significant effect on the selectivity of the process, which bodes well for process intensification. In this paper, we will report the demonstration of the FTR-enabled CE-DKR methodology to the sustainable kinetic resolution of a chiral primary amine at scales up to 100 g—an unprecedented scale reported for the DKR of chiral amines.

## Results and discussion

Building upon our earlier success[48], we sought to further validate the potential for this methodology to synthesize a range of industrially relevant compounds, focusing on compounds synthesized by the ChiPros® process, as well as other bioactive building blocks. Given that the kinetic resolution is enzyme-specific, only the generality of the FTR by Pd/γ-Al$_2$O$_3$ is presented here (Table 2 and Section S2.1.1 in the Supplementary Info). As temperature was found to be the dominant effect[49], the racemization of each substrate was performed at 140, 185, and 230 °C, while the residence time was held constant (9 s).

The overall trend is in keeping with previous reports of Pd racemization catalysts[21–25]: For the racemization of primary benzylic substrates **1a-1f** (entries 1–6), both electron-donating and withdrawing substituents can be accommodated, with the exception of chloride (entries 4 and 5), where no racemization was observed, even at 230 °C. The racemization is enhanced by introducing an N-substituent (entry 1 vs entry 8), but the racemization of the tertiary amine **1i** was much slower (entry 9), and the aliphatic amine **1k** did not racemize under these conditions (entry 11). Last but not least, chiral amines containing pharmacophore-like structures were also tested N-containing hetero- (**1j**) and carbo-cyclic structures (**1m** and **1n**); the latter were found to racemize readily at 140 °C (entries 12–14).

During the exploration of substrate scope, (R)-(+)-1-(3-methoxyphenyl)ethylamine (**1c**) was identified as a suitable candidate for

## Table 2 | Flash-thermal-racemization of optically active amines[a]

| Entry | Substrate | T (°C) | e.e. (%)[b] | Selectivity (%)[c] |
|---|---|---|---|---|
| 1 | (S)-1a | 140 | 41 | 91 |
| 2 | (S)-1b | 140 | 85 | 97 |
| 3 | (S)-1c | 185 | 40 | 83 |
| 4 | (S)-1d | 140 | 100 | 97 |
| 5 | (S)-1e | 230 | 100 | 97 |
| 6 | (S)-1f | 230 | 78 | 90 |
| 7 | (S)-1g | 185 | 94 | 97 |
| 8 | (S)-1h | 140 | 6 | 83 |
| 9 | (S)-1i | 140 | 85 | 95 |
| 10 | (S)-1j | 185 | 84 | 81 |
| 11 | (S)-1k | 230 | 100 | 87 |

## Table 2 (continued) | Flash-thermal-racemization of optically active amines[a]

| Entry | Substrate | T (°C) | e.e. (%)[b] | Selectivity (%)[c] |
|---|---|---|---|---|
| 12 | (S)-1l | 140 | 45 | 80 |
| 13 | (S)-1m | 140 | 20 | 96 |
| 14 | (S)-1n | 140 | 30 | 100 |

[a]Amine substrate (82.5 mM) in anhydrous toluene, γ-Pd/Al$_2$O$_3$ (200 mg), τ = 9 s, 140–230 °C.
[b]Determined by chiral HPLC (Eq. S5, Supplementary Info).
[c]Amount of the original amine remaining after FTR (Eq. S2, Supplementary Info), as determined by GC.

demonstrating the scalability of the FTR-DKR methodology. It is one of the chiral amines currently produced through the ChiPros® process, and serves as a valuable intermediate used in dietary supplements and formulation of personal care products. Moreover, it is a precursor for the synthesis of Tecalcet—an orally active calcimimetic compound used in the treatment of hyperparathyroidism[50].

Previous research into the asymmetric synthesis of **1c** provided useful benchmarks for our work. Notably, Han et al. reported a one-pot batch process incorporating catalytic hydrogenation and chemoenzymatic dynamic kinetic resolution (Scheme 4)[51], whereby Pd/AlO(OH) was combined with a slight H$_2$ pressure to facilitate the reversible reduction of oxime **6** to **1c**. At the same time, CALB catalyzed the enantioselective acylation of (R)-**1c** using methyl 2-methoxylacetate (**3a**), a process that requires 3 days to produce just over 1 g of the chiral amide (**2c**). More recently, Tan et al described an enantioselective reductive amination of the arylmethyl ketone (**7**), catalyzed by a chiral phosphine-ligated ruthenium catalyst under H$_2$ atmosphere[52]. The larger-scale reaction was exemplified by the use of only 750 mg of the ketone, affording 670 mg of (R)-**1c** in 89% yield and 94% e.e. after 1 day. The requirement for a very high H$_2$ pressure (57 bar), alongside the use of toxic and environmentally hazardous fluorinated solvent (TFE), presents significant challenges for the translation of this methodology into a commercially viable process.

In our preliminary investigation, a total of 18 heterogeneous transition metal catalysts were systematically evaluated for their efficacy in racemizing (S)-**1a**, using a screening deck reactor (Section S.2.1.2 of the Supplementary Info). The results of this catalyst screening indicated that supported palladium (Pd) and platinum (Pt) catalysts exhibited particularly high levels of catalytic activity. Following this initial assessment, these two catalyst types were further investigated for their ability to racemize (R)-**1c**, and the findings are presented in Table 3.

Among the three Pd catalysts evaluated (Table 3, entries 1–3), the highest degree of racemization, indicated by the lowest enantiomeric excess (e.e.), was observed with both Pd(OH)$_2$/C (Pearlman's catalyst) and Pd/C. In contrast, the Pd/γ-Al$_2$O$_3$ catalyst was able to suppress side product formation much more effectively (100% selectivity), compared to its carbon-supported counterparts (entries 2 and 3 vs. entry 1). The acid-base properties of Pd catalysts were reported to significantly influence both the rate and selectivity of the racemization of chiral amines[52]. Hence, the enhanced activity of Pd(OH)$_2$/C can be attributed to the basic characteristics of PdO, which promote the formation of surface amide intermediates (Scheme 5).

**Scheme 4 | Previous (near-) gram-scale asymmetric synthesis of (R)-1c (Ar=3-MeOC$_6$H$_4$). A** One-pot chemoenzymatic dynamic kinetic resolution of an oxime. **B** Catalytic asymmetric reductive amination of a ketone.

(A) Han et al (2010):

(B) Tan et al (2018):

## Table 3 | Selected Pd and Pt catalysts for the racemization of (R)-1c (Scheme 5)$^a$

| Entry | Catalyst | e.e.(%)$^b$ | Selectivity(%)$^c$ |
|-------|----------|-------------|---------------------|
| 1 | Pd/γ-Al$_2$O$_3$ | 68 | 100 |
| 2 | Pd/C | 53 | 44 |
| 3 | Pd(OH)$_2$/C | 29 | 47 |
| 4 | Pt/γ-Al$_2$O$_3$ | 83 | 95 |
| 5 | Pt/C | 80 | 72 |

$^a$Reactions were conducted in duplicate in parallel using a Deck Screening Pressure Reactor: Amine (82.5 mM in toluene), 145 °C, 1 h, 1 mol% catalyst.
$^b$Enantiomeric excess. Determined by chiral HPLC.
$^c$Amount the **1c** remaining after FTR (Eq. S2, Supplementary Info). Determined by HPLC.

Conversely, the observed low selectivity with the catalyst supported on carbon (*ca.* 45%) may be linked to the high surface area and effective adsorption properties of activated carbon, which retain imine and amide intermediates on the surface, thereby facilitating intermolecular reactions that yield undesirable side products.

In comparison, the racemization of amines over Pt catalysts occurred at a considerably slower rate (entries 4 and 5). Once again, the selectivity was notably inferior when employing carbon support in contrast to alumina. From this limited catalyst screening (Table 3), we can surmise that the redox properties of the metal nanoparticles play a critical role in determining the extent of racemization (e.e.), while the nature of the support significantly influences the selectivity outcomes.

The superior selectivity demonstrated by Pd/γ-Al$_2$O$_3$ led us to select it as the preferred catalyst for the racemization of (R)-**1c**. In contrast to our earlier studies conducted using dilute amine solutions (<100 mM)[48,49], the FTR study in this work was performed using a synthetically useful amine concentration (0.25–1 M). Subsequently, a DoE model consisting of a 25-experiment I-optimal split-plot screen was generated to examine the racemization of (R)-**1c** (0.25–1 M) between 120 and 240 °C and residence times of between 4 and 20 s (Fig. 1 and Supplementary Table 3 of the Supplementary Info).

The findings from the DoE study closely mirror our previously conducted studies. Specifically, selectivity can be enhanced by reducing the residence time, effectively suppressing the formation of side products, whereas variations in temperature and amine concentration did not yield any significant impact on selectivity. Conversely, the racemization of (R)-**1c** (e.e.) displayed a strong dependence on temperature and flow rate, whereas amine concentration appeared to exert no influence on this process. Such "pseudo-zero order" observed for the amine may be ascribed to the small amount of accessible active catalytic sites in the packed bed (only 200 mg of the catalyst was used) compared to the large quantity of amine (0.25–1 M) in a single pass, i.e., the adsorption of the system is saturated.

To check for catalyst deactivation, selected experiments from the initial set of twenty-five were repeated both within and in addition to the DoE experiments (Supplementary Info Table 4, Supplementary Info). The results

showed very good reproducibility, proving that Pd catalyst remained stable over the course of these experiments. The DoE model predicted a potential optimum for the racemization of (R)-**1c** at 230 °C in 4 s, to afford **1c** with 41% e.e. and 94% selectivity. Pleasingly, this aligned extremely well with experimental results, where **1c** was obtained in 38% e.e. and 91% selectivity under the stated conditions.

Most significantly, the DoE study suggests that it is possible to utilize a highly concentrated solution of amine at high flow rates (short residence time) with no detectable catalyst deactivation, which are important criteria for process intensification. Encouraged by these results, the racemization of a 1 M solution of (R)–**1c** was performed on a larger scale (15 g) in a batch-recycle process (Fig. 2). Over 80 min, the enantiomeric excess of the amine was reduced to 4% e.e. with an overall selectivity of 80% (blue datapoints). Using the same amount of the Pd catalyst (200 mg), the process was subsequently scaled up ten-fold to 150 g, whereupon the e.e. of (R)-**1c** can be reduced to 6% in 15 h (orange data points), and the primary amine can be recovered in 82% yield.

Next, the kinetic resolution of (*rac*)-**1c** by methyl 2-methoxyacetate (**3a**) catalyzed by Novozym-435 in a packed-bed reactor was also optimized using DoE. Setting the flow rate at 5 mL min$^{-1}$ to align with the racemization step, the effect of temperature (20–70 °C), acyl donor equivalents (0.5–2) and amine concentration (0.25–1 M) on conversion and e.e. were explored (Fig. 3 and Table S5 of Supplementary Info). The model created showed a high level of precision and demonstrated that 42% conversion could be achieved in a single pass using 1 M of (*rac*)-**1c**, two equivalents of **3a** at 70 °C, affording the kinetically resolved amide product **1c** with excellent optical purity (99% e.e.).

Finally, the two PBRs were combined in tandem in a recycle-batch system, where the reactants were recycled continuously *via* a reservoir (Fig. 4 and Section 4.4 in Supplementary Info). Previously, molecular sieves were deployed to remove the methanol byproduct which can cause deactivation of the catalyst. To cater for the increase in reaction scale, the molecular sieves were replaced by a distillation column fitted to the reaction reservoir, which was heated at 70 °C during the reaction, to extract methanol from the reaction mixture continuously (methanol forms an azeotrope with toluene at 64 °C, with an approximate 12% toluene content).

This system was used initially tested using 10 g of (*rac*)-**1c**. We were delighted to see that the system was able to convert 91% of the amine to the desired (R)-**2c** with 96% e.e. and 84% selectivity in 2 h and 20 min. Encouraged by this, the process was attempted on a 100 g scale. For operational reasons, the reaction was conducted over 4 days only during working hours; during which aliquots from the reservoir were extracted every hour and analyzed, before the system was shut down overnight (indicated by gray vertical lines in Fig. 5). Despite these interruptions, catalytic turnovers were able to resume after each scheduled downtime. After 14 hours of operation, the reaction proceeded to 95% conversion with excellent retention of the optical purity of the amide (99% e.e.). However, a noticeable decrease in the selectivity for the primary amine was detectable, decreasing to 96% after *ca.* 5 h, and 88% after 8 h. We attribute this erosion of selectivity to the presence of high concentration of the amide product in

**Scheme 5 | Proposed mechanism for racemization and side product formation on catalyst surfaces. The role of support materials on product selectivity.**

**Fig. 1 | The 3-factor, 2-responses DoE optimization of the FTR of (R)–1c over Pd/γ-Al₂O₃.** Flow rates of 1–5 mL min⁻¹ correspond to residence times of 4–20 s.

the reaction mixture (*ca*. 60% at 5 h), which may competitively bind with the enzyme active site, reducing the rate of kinetic resolution, allowing the formation of side products to become competitive.

In an authoritative review on industrial biocatalysis by Bornscheuer and co-workers[53], (recently updated[54]) the range of STYs of industrial processes was reported between the range of 0.001–0.3 kg L⁻¹ h⁻¹, and values exceeding 10 g L⁻¹ h⁻¹ are rare. Using this as a benchmark, the productivity values of known literature reports of CE-DKR of chiral amines, performed on ≥1 g of the racemic precursor, are collated in Table 4. Once again, to ensure fair comparison between "one-pot" (entries 2, 3, and 5) and recycle-batch processes, the total amount of solvents deployed in the process was deployed as the volume element in the calculation of STY's (Eq. 1), rather than the size of the reactor, which will intrinsically favor the batch-recycle system.

The calculations show that the FTR methodology delivered a STY of 73.2 g L⁻¹ h⁻¹ for the reaction performed with 10 g of the amine (Table 4,

entry 6), which is the highest reported to date for the CE-DKR of chiral amines. Using the same amounts of catalysts (200 mg and 3 g of the chemo- and bio-catalysts, respectively) for the 10-fold scaleup, the STY decreases to 13.6 g L⁻¹ h⁻¹ (entry 7) as may be expected, but still well within the productivity expected of an industrial scale process, without compromising the optical purity of the resolved amide (99% ee). While this may be lower than the STY calculated for the one-pot synthesis of a different chiral amine **10** reported by Jia (entry 5), it is important to note that the amounts of catalysts deployed by the FTR protocol were also substantially lower than the one-pot procedure (200 mg vs 27.5 g for Pd, and 3 g vs 18.3 g for the biocatalyst). Given that the price of precious metal catalysts is a significant contributor to the cost of goods, this result can be considered as a milestone for chemoenzymatic kinetic dynamic resolution of chiral amines.

Last but not least, the Green Chemistry metrics of these gram-scale reactions were compared using the CHEM21 toolkit, including atom economy (AE), reaction mass efficiency (RME) and process mass intensity

## Table 4 | Comparison of scale (≥1 g) and productivity of chemoenzymatic dynamic kinetic resolution of chiral amines

| Entry | Resolved amide | Yield (%)[a] | e.e. (%)[b] | Scale (g)[c] | Time (h) | STY ($g_{amide}$ $L^{-1}$ $h^{-1}$) | Catalysts (quantity) | Ref. |
|---|---|---|---|---|---|---|---|---|
| 1 | (R)-2a | 89 | 95 | 1 | 1 | 1.42 | 5% Pd/γ-Al$_2$O$_3$ (200 mg) Novozym-435 (1.7 g) | Hii[48] |
| 2 | (R)-2a | 83[d] | 98 | 6.8 | 24 + 72[e] | 0.32 | Ar = 4-MeOC$_6$H$_4$ (745 mg) Novozym 435 (340 mg) | Bäckvall[56] |
| 3 | (R)-8 | 81.6[d] | 96 | 3 | 24 × 2[e] | 2.40 | (34.3 mg) Candida Rugosa (2.1 g) 5% Pd/γ-Al$_2$O$_3$ (200 mg), Novozym 435 (1.7 g) | Blacker[30] |
| 4 | (R)-2m | 92 | 99 | 1 | 1 | 15.4 | 5% Pd/γ-Al$_2$O$_3$ (200 mg), Novozym 435 (1.7 g) | Hii[48] |
| 5 | (R)-2m | 80.1[d] | >99 | 73.2 | 12 | 20.6 | 5% Pd/AlOOH (27.5 g) CALB (18.3 g) | Jia[57] |
| 6 | (R)-2c | 76 | 96 | 10 | 2.3 | 73.2 | 5% Pd/γ-Al$_2$O$_3$ (200 mg), Novozym 435 (3 g) | This work |
| 7 | (R)-2c | 84 | 99 | 100 | 14 | 13.3 | 5% Pd/γ-Al$_2$O$_3$ (200 mg), Novozym 435 (3 g) | This work |

[a]Unless otherwise stated, this was based on the amount of amine converted into amide (conversion × selectivity).
[b]Amount of (rac)-amine employed in the reaction.
[c]Enantiomeric excess. Determined by chiral HPLC.
[d]Amount of (racemic) amine utilized in the reaction.
[e]Recharge of catalyst (with or without additional acylating agent) was required during the reaction.

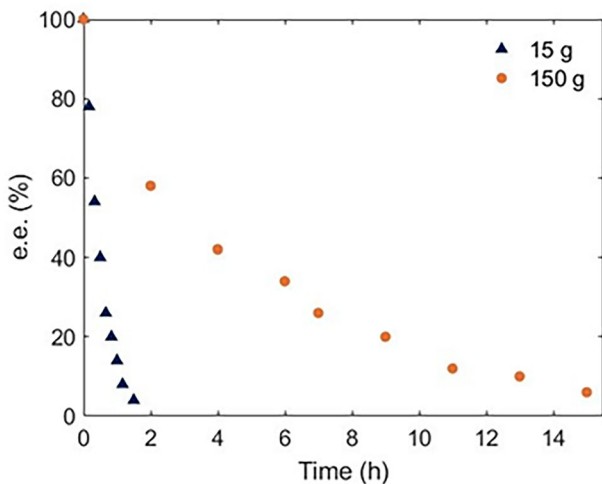

**Fig. 2 | FTR of (R)-1c over Pd/γ-Al2O3 at different scales.** 15 g (blue triangle) and 150 g (orange circle).

(PMI) (Table 5)[55]. Comparing the result from this work (entries 6 and 7) from our preliminary work (entries 1 and 3), improvements across all the metrics have been achieved. What is particularly notable is the PMI value of <7, which is among the lowest recorded for this process.

## Conclusions

In conclusion, we have shown that the flash-thermal racemization protocol can be successfully scaled up to 100 g and demonstrated that the FTR methodology outperforms previously reported catalytic systems in a quantifiable way, in terms of productivity, green chemistry metrics, and cost-effectiveness, setting a new milestone for the industrial production of optically active amines by CE-DKR methods. However, the study also highlighted areas where further development work is needed; namely, reduced selectivity at high amide conversion. Potential amide inhibition may be addressed by modifying the reactor design to separate the optically active amide from the unresolved amine following kinetic resolution, for example. Work is currently underway to utilize the data collected in this work to design different reactor systems, as well as technoeconomic and sustainability assessments, which we hope to report in due course.

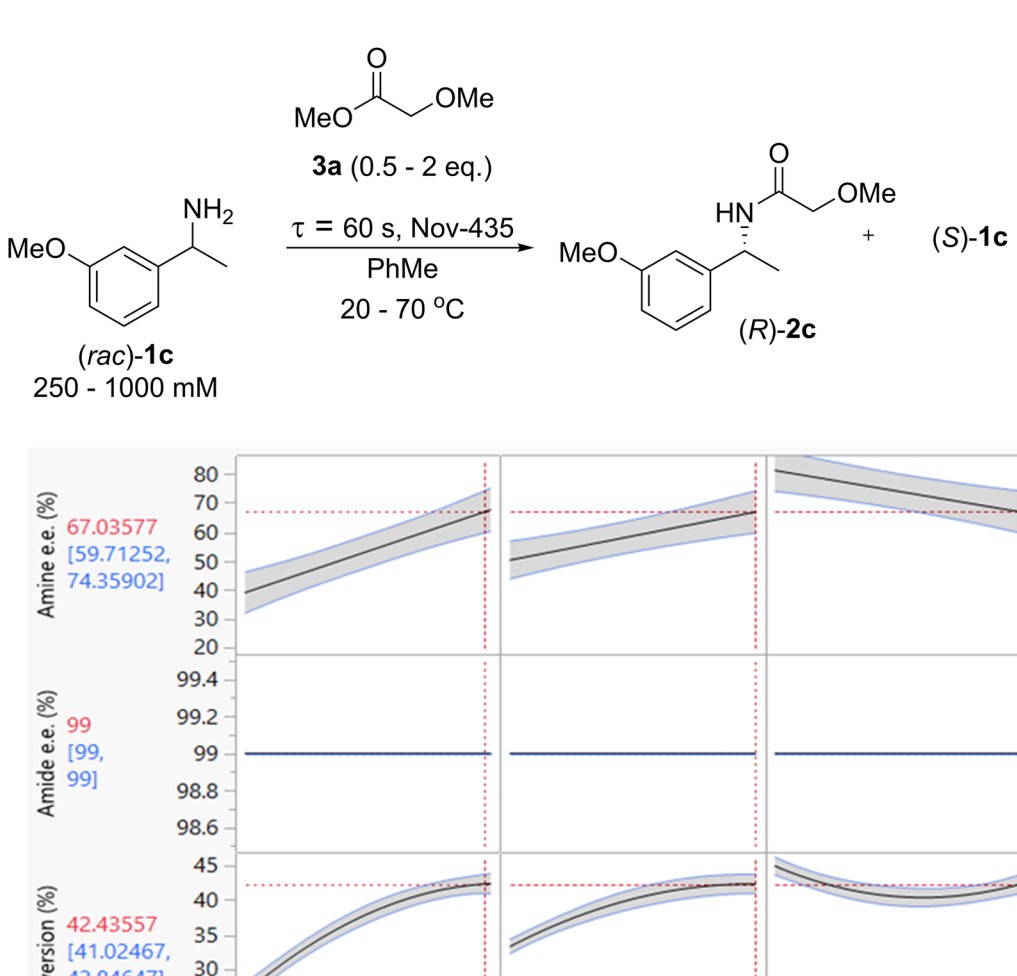

**Fig. 3 | The 3-factor, 2-responses DoE optimization of the KR of (rac)-1c with Novozym-435.** Flow rate set at 5 mL min⁻¹.

**Fig. 4 | FTR-CE-DKR flow system.** Reaction conditions: (rac)-1c (1 M in toluene), 5% Pd/γ-Al₂O₃ (200 mg, 230 °C), and Novozym 435 (3 g, 70 °C), flow rate = 5 mL min⁻¹.

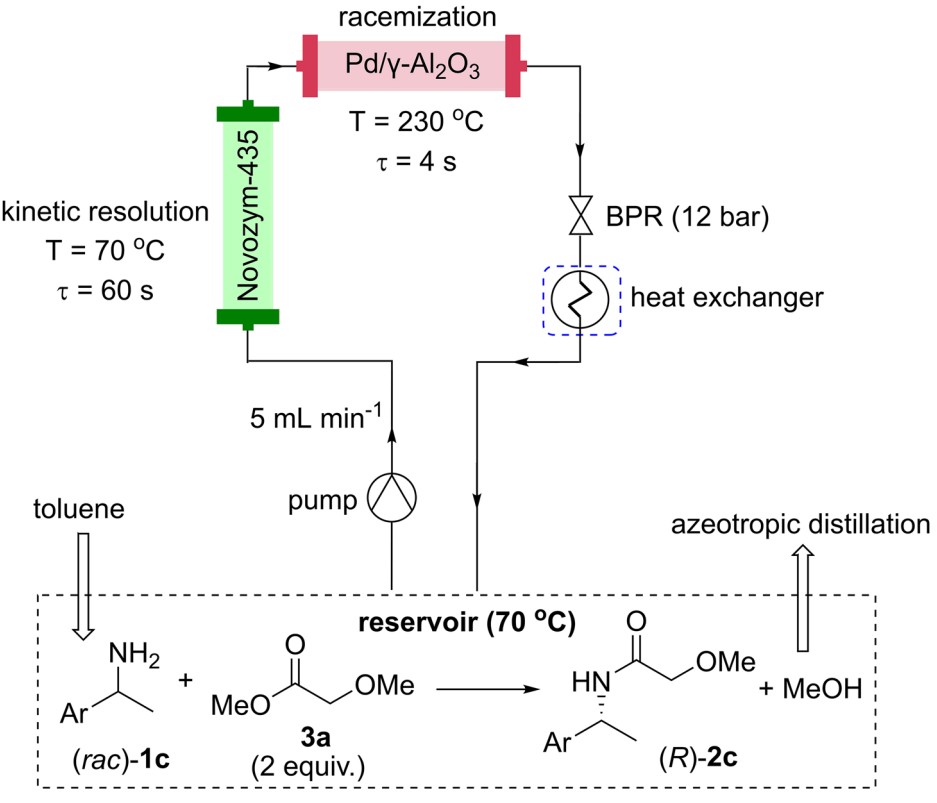

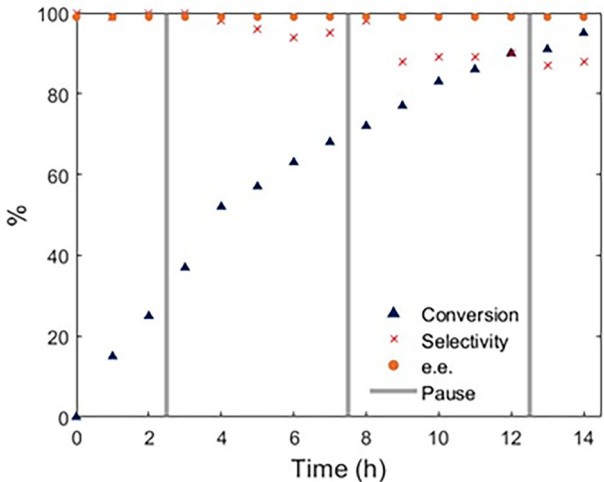

**Fig. 5 | The reaction progress of the 100 g CE-DKR of (rac)-1c to (R)-2c.** The gray bars mark overnight shutdowns.

**Table 5 | Comparisons of green chemistry metrics of CE-DKR of chiral amines performed at ≥1 g scale**

| Entry | Amine | Scale (g) | AE[a] | RME[b] | PMI[c] | Ref. |
|-------|-------|-----------|-------|--------|--------|------|
| 1 | **1a** | 1 | 80.8 | 10.8 | 73.6 | Hii[48] |
| 2 | **1a** | 10 | 80.8 | 63.5 | 29.8 | Bäckvall[56] |
| 3 | **1m** | 1 | 76.9 | 44.0 | 68.4 | Hii[48] |
| 4 | **1m** | 72 | 72.7 | 49.6 | 6.2 | Jia[57] |
| 5 | **1c** | 1 | 67.8 | 47.4 | 69.3 | Kim[51] |
| 6 | **1c** | 10 | 87.5 | 47.5 | 7.5 | This work |
| 7 | **1c** | 100 | 87.5 | 59.0 | 5.8 (6.9)[d] | This work |

[a]Atom economy, AE = (mwt of product)/(sum of mwt of all reactants). Higher AE means less waste.
[b]Reaction Mass efficiency, RME = (mass of products)/(total mass of all reactants). A higher RME indicates a more cost-effective process.
[c]Process Mass Intensity, PMI = (total mass of all input materials, including solvents)/(mass of product). Lower PMI indicates a more efficient process. Unless otherwise indicated, the PMI values were calculated without including materials and solvents used in the workup procedure, as these are usually not included in earlier papers.
[d]Including acid workup.

## Methods

### 100 g scale FTR-CE-DKR of R–2-methoxy-N-(1-(3-methoxyphenyl)ethyl)acetamide, (R)-2c

The reactor was primed by anhydrous toluene at 1 mL min⁻¹ for 10 min, before the KR and FTR reactors were heated to 70 °C and 230 °C, respectively. Once the desired temperatures were reached, the inlet was switched to the delivery a solution of (rac)–1c (100 g, 661 mmol) and methyl 2-methoxyaceate, 3a (138 g, 2 equiv.) in toluene (661 mL, 1 M) from a 1 L, 3-necked RB flask fitted with a distillation column. During the reaction, the RB flask was heated and stirred at 70 °C to remove the MeOH by-product *via* azeotropic distillation. After 14 h, the reaction reached 95% conversion and

90% selectivity (HPLC). The reaction mixture was cooled down, washed with 2 M HCl (3 × 50 mL), dried with MgSO₄, filtered, and reduced in vacuo to yield (R)-2c as an orange oil (101 g, 68%, 99% e.e.).

## Data availability

Data generated during the study are either included in the Supplementary Information or available from the corresponding author upon reasonable request.

**Article**

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

## Acknowledgements

We are grateful to Dr Benjamin Deadman (BJ Deadman Consultancy) for advice on the DoE work. The work was funded by an EPSRC Prosperity Partnership grant: Innovative Continuous Manufacturing for Industrial Chemicals ("IConIC", EP/X025292/1), with additional cash contributions from BASF SE.

## Author contributions

The experiments were performed by M.J.T., supervised by K.K.M.H. and K.H., while the other authors (D.M.M., P.S., J.D., and C.H.) provided materials, advice, and direction during the project. The manuscript was written by M.J.T. and K.K.M.H., and all authors have given approval to the final version of the manuscript.

## Competing interests

The authors declare no competing interests.
