## [Transparent Peer Review file · Communications Chemistry]

Scalable and Sustainable Synthesis of Chiral Amines by Biocatalysis

Corresponding Author: Professor King Kuok Hii

Version 0:

Reviewer comments:

Reviewer #1

(Remarks to the Author)

The manuscript describes the scale-up of a palladium-catalyzed flash-thermal racemization protocol for (S)-1-(3-methoxyphenyl)ethylamine, used in combination with a biocatalytic enantio-discriminating acylation with methyl methoxyacetate. This process is a key step in the synthesis of (R)-1-(3-methoxyphenyl)ethylamine, which is a valuable intermediate for supplements and pharmaceuticals. Although I'm not familiar with chemical engineering, I found the manuscript to be well-written and clearly organized. The protocol was successfully demonstrated on a 100 g scale. However, I have several concerns regarding the relevance of the current work to other reported methods and the author's previous works. In Tables 3 and 4, the authors compare the current protocol with previous methods that used different substrates. I believe such comparisons are of limited value. It would strengthen the manuscript if the author could demonstrate the applicability of this protocol to other substrates. Have the author tested current protocol with other substrates? Additionally, since the reaction system itself has already been reported by the authors (Ref. 48 and 49), and the present work focuses on further optimization for scale-up, I am somewhat skeptical about the novelty of the study. That said, as mentioned above, I am not an expert in engineering, and thus I would not fully assess the extent of advancement over the previous works. Finally, in Figure 7, there appears to be a sudden decrease in selectivity after 8–9 hours. Could the authors explain the reason for this sudden drop? Overall, I consider the manuscript suitable for publication after minor revisions addressing the points raised above.

Reviewer #2

(Remarks to the Author)

The manuscript reports a technically sound and thoughtfully executed study that builds upon the authors' previous work on Flash Thermal Racemization (FTR) applied to dynamic kinetic resolution (DKR) of chiral amines. The integration of catalyst screening, DoE-based optimization, and scale-up to a gram process provides valuable insights and represents a significant improvement over conventional approaches, especially in terms of productivity and sustainability metrics. The reported STY values and green chemistry indicators are noteworthy and align well with the principles of process intensification and greener manufacturing.

However, while the work demonstrates incremental innovation and industrial relevance, it heavily relies on concepts and methodologies already established by the authors in prior publications. The novelty, therefore, appears more evolutionary than revolutionary, which may limit its urgency for publication in *Communications Chemistry*, a journal that typically seeks breakthrough advances.

Furthermore, although the manuscript provides detailed productivity and CHEM21 green chemistry metrics, it is not clearly stated whether these calculations include product isolation steps. The absence of explicit information regarding the isolation and purification of the resolved amide raises concerns about the real-world applicability and comparability of the reported process metrics. Clarifying whether product isolation was included in PMI and RME calculations would significantly strengthen the manuscript's credibility and transparency.

In summary, the manuscript presents compelling results in the context of chiral amine resolution and process sustainability, but would benefit from a clearer positioning of its novelty and more transparent reporting of the isolation procedures and their impact on sustainability metrics.

Reviewer #3

(Remarks to the Author)

Takle et al. report the development of a procedure for flash thermal racemization in the chemoenzymatic dynamic kinetic resolution (FTR-CE-DKR) of an optically active chiral primary amide. This work builds on previous studies (refs. 48–49) in which they disclosed a novel method of chemoenzymatic DKR in a flow system, developing the ‘flash thermal racemization’ (FTR) conditions without extraneous additives. They further explored the variables of temperature, amine concentration, and flow rate using DoE to achieve optimal ee and selectivity. These earlier studies represented a key technological advancement to the long-established ChiPros® process, developed and used by BASF for over 20 years in the manufacture of optically active amines—including important bulk herbicide intermediates—at the multi-kiloton scale. Specifically, the authors coupled a metal-catalyzed racemization step with the lipase-mediated kinetic resolution to achieve a dynamic kinetic resolution, thereby avoiding multiple operations (including energy-intensive distillations for isolation and recovery of amines) required in the ChiPros® process. Although the present work may appear incremental, its true breakthrough lies in the successful screening and optimization studies, together with the scale-up of the biocatalytic reaction to the 100 g scale. This development provides a compelling proof of concept for the industrial application of biocatalysis in the production of optically active amines. The study is comprehensive, clearly written, and includes sufficient experimental detail to enable reproducibility. Some adjustments are needed:

154–156: “Based on empirical observations made in our earlier work, 5 commercially available Pd and Pt catalysts were chosen for this study (Table 2).” → Suggest clarifying the selection criteria in more detail.

164 Table 2: The header “e.e.[b]/%” is confusing. Recommend using “e.e. (%)” and moving [b] (“determined by chiral HPLC”) to the footnote. Ensure consistency with Figure 4 Y-axis.

166 [c]: “Retention of primary amine moiety. Determined by GC.” → Clarify how selectivity is calculated. Is it the % of amine in the reaction mixture compared with byproducts 4c and 5c? Is this based on a calibration curve? Add explanation in the main text.

195: “Residence times of between 4–20 s (Scheme 7 and Table S3, Supplementary Info)” → Table S3 reports DoE results for kinetic resolution, not residence times. Please clarify earlier (line 218) and use consistent terminology. Consider moving the explanation from SI lines 62–63 into the main text (e.g., footnote to Figure 3 or inline).

210–211: “were repeated both within, and in addition to, the DoE experiments (Table S4, Supplementary Info).” → Table S4 is missing from the SI.

222–223: Revise to: “subsequently scaled up ten-fold to 150 g, whereupon the e.e. of (R)-1c can be reduced to 6% in 15 hours (Fig. 4, red data points), and the primary amine can be recovered in 82% yield.”

232–233: “equivalents (0.5–2) and amine concentration (0.25–1 M) were explored (Scheme 8, Fig. 5 and Table S5, Supplementary Info).” → This is Table S3, not S5.

Figure 3: Format consistently as “°C.”

Figure 4: Label axes consistently: Y = “e.e. (%)”, X = “Time (min)”.

Figure 5: Add units of measure consistently, as in Figure 3.

278–279: “Using the same amount of catalysts (200 mg and 3 g of the chemo- and biocatalysts, respectively) for the 10-fold scale-up, the STY decreases to 13.6 g L⁻¹ h⁻¹ (entry 7) as may be expected, but still well within the productivity expected of an industrial...” → The difference between entries 6 and 7 is unclear. Add a clarifying footnote in Table 3.

Table 3 header: Change “Quantity” to “mg or g.”

Version 1:

Reviewer comments:

Reviewer #2

(Remarks to the Author)

After corrections made by the authors I believe the manuscript is ready for publication.

Point-by-point response

We are grateful to the reviewers for taking their time to review the manuscript and making valuable suggestions. We have carefully considered each of the points made, and point-by-point responses are provided below, along with changes we have made in the revised manuscript.

Reviewer #1 (Remarks to the Author):

The manuscript describes the scale-up of a palladium-catalyzed flash-thermal racemization protocol for (S)-1-(3-methoxyphenyl)ethylamine, used in combination with a biocatalytic enantio-discriminating acylation with methyl methoxyacetate. This process is a key step in the synthesis of (R)-1-(3-methoxyphenyl)ethylamine, which is a valuable intermediate for supplements and pharmaceuticals. Although I'm not familiar with chemical engineering, I found the manuscript to be well-written and clearly organized. The protocol was successfully demonstrated on a 100 g scale. However, I have several concerns regarding the relevance of the current work to other reported methods and the author's previous works.

In Tables 3 and 4, the authors compare the current protocol with previous methods that used different substrates. I believe such comparisons are of limited value. It would strengthen the manuscript if the author could demonstrate the applicability of this protocol to other substrates. Have the author tested current protocol with other substrates?

Response: We are pleased that the reviewer appreciate the clarity of the manuscript. While the preliminary work had been published, the present manuscript focusses on the of the FTR methodology at scale, using a different chiral amine target that is of commercial interest. In the original paper, while we were able to show gram-scale synthesis and good productivity, the sustainability metrics (as measured by PMI) is too low. Hence this manuscript focuses on process intensification aspects, leading to a an improvement in sustainability metrics, which is often overlooked in other manuscript on CE-DKR, but nevertheless critical for technoeconomic and life cycle assessments.

Nevertheless, per the reviewer's request, we have included a substrate scope study in the revised manuscript (Table 2, and a short discussion). These does contain some novel substrates (particularly the pharmacophores) which have not been previously reported.

Additionally, since the reaction system itself has already been reported by the authors (Ref. 48 and 49), and the present work focuses on further optimization for scale-up, I am somewhat skeptical about the novelty of the study. That said, as mentioned above, I am not an expert in engineering, and thus I would not fully assess the extent of advancement over the previous works.

Response: Our preliminary paper (ACS Catal 2023) focussed on the basic principles/design of the FTR methodology, demonstrated on a 1 g-scale. The main focus of the current paper is on the demonstration of the scalability of the methodology, as well as the associated sustainability metrics, including substantial reduction in solvent use, and removing the need for a solid dessicant. We are able to demonstrate, for the first time, that the FTR-DKR of chiral amine can be achieved on a 100g scale, not only with unprecedented productivity (STY values commensurate with industrial scale biocatalytic processes), but also in a more sustainable manner (favourable AE, RME and PMI values) compared to previous reports. While there is no new molecular structures or transformations reported in this work, this is nevertheless a significant milestone for the realisation of DKR methodologies for industrial applications (as recognised by Reviewer #3 below). At the same time, it also highlights further issues and insights, e.g. catalyst deactivation, to be addressed in future designs.

Finally, in Figure 7, there appears to be a sudden decrease in selectivity after 8–9 hours. Could the authors explain the reason for this sudden drop?

Response: Indeed, the selectivity started to drop after 5 h, and more dramatically after the 8 hour mark, after the second shut-down-and-restart cycle. We have addressed this in the text, where we attribute this to the very high conversion and high concentration of amide products present, which competitively bind with the enzyme active site. This prevents the amine from binding, thus making them more vulnerable to competitive side reactions.

Reviewer #2 (Remarks to the Author):

The manuscript reports a technically sound and thoughtfully executed study that builds upon the authors' previous work on Flash Thermal Racemization (FTR) applied to dynamic kinetic resolution (DKR) of chiral amines. The integration of catalyst screening, DoE-based optimization, and scale-up to a gram process provides valuable insights and represents a significant improvement over conventional approaches, especially in terms of productivity and sustainability metrics. The reported STY values and green chemistry indicators are noteworthy and align well with the principles of process intensification and greener manufacturing.

However, while the work demonstrates incremental innovation and industrial relevance, it heavily relies on concepts and methodologies already established by the authors in prior publications. The novelty, therefore, appears more evolutionary than revolutionary, which may limit its urgency for publication in Communications Chemistry, a journal that typically seeks breakthrough advances.

Response: We acknowledge that the manuscript does not describe any new molecules or transformations. Normally, such standalone 'scale-up' work can be considered as incremental. However, as stated above, the focus of the manuscript is on the process intensification and improvement in sustainability metrics, resulting in quantifiable results that represent a major breakthrough in the production of chiral amines not only in terms of productivity, but also better sustainable metrics. Such scalable and sustainable processes deserve greater attention in the 21st Century.

Furthermore, although the manuscript provides detailed productivity and CHEM21 green chemistry metrics, it is not clearly stated whether these calculations include product isolation steps. The absence of explicit information regarding the isolation and purification of the resolved amide raises concerns about the real-world applicability and comparability of the reported process metrics. Clarifying whether product isolation was included in PMI and RME calculations would significantly strengthen the manuscript's credibility and transparency.

Response: This is very good point. However, it is not always possible to compare apples to apples, as not all workup procedure (if reported at all) contain details on the amount of solvents used in the extraction or purifications, making direct comparisons of PMI particularly difficult. We have clarified this in the footnote [c] to Table 4, and also providing an excel spreadsheet showing how these numbers are derived, with accompanying experimental procedures, so that the reader can make their own decisions. We have also included the PMI calculation including workup for our work (entry 7, footnote [d])

Reviewer #3 (Remarks to the Author):

The study is comprehensive, clearly written, and includes sufficient experimental detail to enable reproducibility.

Response: We are grateful to the reviewer for the complimentary remarks.

154–156: “Based on empirical observations made in our earlier work, 5 commercially available Pd and Pt catalysts were chosen for this study (Table 2).” → Suggest clarifying the selection criteria in more detail.

Response: We have added our earlier screening study (conducted with amine substrate 1a) in the supplementary info (Table S2)

164 Table 2: The header “e.e.[b]/%” is confusing. Recommend using “e.e. (%)” and moving [b] (“determined by chiral HPLC”) to the footnote. Ensure consistency with Figure 4 Y-axis.

Response: The headers for all tables and figures have been altered to reflect these comments and highlighted in the manuscript.

166 [c]: “Retention of primary amine moiety. Determined by GC.” → Clarify how selectivity is calculated. Is it the % of amine in the reaction mixture compared with byproducts 4c and 5c? Is this based on a calibration curve? Add explanation in the main text.

Response: We apologize for the mistake – GC was not utilize in this work. All conversions were determined using uPLC and anisole as internal standard. The details are given in the Supplementary Info (Section S1)

195: “Residence times of between 4–20 s (Scheme 7 and Table S3, Supplementary Info)” → Table S3 reports DoE results for kinetic resolution, not residence times. Please clarify earlier (line 218) and use consistent terminology. Consider moving the explanation from SI lines 62–63 into the main text (e.g., footnote to Figure 3 or inline).

Response: We agree with the reviewers comments, and have clarified the residence time achieved by each flow rate has been added to the footnote of Figure 3, as suggested.

210–211: “were repeated both within, and in addition to, the DoE experiments (Table S4, Supplementary Info).” → Table S4 is missing from the SI.

Response: We thank the reviewer for bringing this to our attention. The numbering in the manuscript has been updated.

222–223: Revise to: “subsequently scaled up ten-fold to 150 g, whereupon the e.e. of (R)-1c can be reduced to 6% in 15 hours (Fig. 4, red data points), and the primary amine can be recovered in 82% yield.”

Response: We agree with the reviewers comments. The recommended changes have been made and highlighted in the manuscript.

232–233: “equivalents (0.5–2) and amine concentration (0.25–1 M) were explored (Scheme 8, Fig. 5 and Table S5, Supplementary Info).” → This is Table S3, not S5.

Response: We thank the reviewer for bringing this to our attention. The numbering in the manuscript has been updated.

Figure 3: Format consistently as “°C.”

Response: We agree with the reviewers comments. The recommended changes have been made.

Figure 4: Label axes consistently: Y = “e.e. (%)”, X = “Time (min)”.

Response: We agree with the reviewers comments. The recommended changes have been made.

Figure 5: Add units of measure consistently, as in Figure 3.

Response: We agree with the reviewers comments. The recommended changes have been made

278–279: “Using the same amount of catalysts (200 mg and 3 g of the chemo- and biocatalysts, respectively) for the 10-fold scale-up, the STY decreases to $13.6 \text{ g L}^{-1} \text{ h}^{-1}$ (entry 7) as may be expected, but still well within the productivity expected of an industrial...”
→ The difference between entries 6 and 7 is unclear. Add a clarifying footnote in Table 3.

Response: We thank the reviewer for their comments. The scale of the reaction is indicated in column 5. Footnote [c] is added to further clarify.

Table 3 header: Change “Quantity” to “mg or g.”

Response: We agree with the reviewers comment and have amended the manuscript accordingly.

Additional changes implemented during revision:

1. Reworking of certain paragraphs to improve clarity
2. Revision of compound numbers, as well as number of schemes and figures
3. Insertion of data availability statement
4. Additional excel spreadsheet submitted as supplementary info to support calculations shown in Table 4.
5. Minor typographical and grammatical errors

All changes are highlight in yellow.

Requests from the Editor:

1. In line with our editorial policy, colon is not allowed to present in the title. Please provide an alternative title without colon if you don't agree with the original editor's suggestion.

Response: The precise phrasing of the guidance document states: "Titles do not normally include numbers, acronyms, abbreviations or punctuation.", which suggests that a colon (punctuation) can be accommodated. However, if this really cannot be accommodated, the alternate title for the manuscript is: "Scalable and Sustainable Synthesis of Chiral Amines by Biocatalysis."

2. Please add one more sentence after the figure and scheme title, to describe the figure and scheme in detail.

Response: Done.

3. In line with our editorial policy, the two-sentence summary should contain the first sentence to describe the background and context and the scientific question, then the second sentence to summarize the main conclusion of the paper. Your proposed summary is missing the first contextualized sentence. Please revise and provide an alternative two-sentence summary if you don't agree with the original editor's suggestion.

Response: Done

Requests from the Editorial Assistant:

1. Each complete figure must be supplied at 300 dpi or higher resolution to ensure the highest-quality reproduction in the journal. Please submit your figures following the guidelines: <https://www.nature.com/commsbio/submit/submission-guidelines#figures-publication>

Response: All the figures and schemes are already provided as TIF files with 600 dpi resolution. Note that a new Scheme 4 is provided.

2. Please incorporate funding sources under Acknowledgement section.

Response: Done

3. We notice that there are authors missing from your Author Contributions statement [David M. Maurer, Philipp Staehle, Joachim Dickhaut, Christian Holtze]. Please ensure that all authors are included. Where multiple authors possess identical initials, they must be clearly disambiguated from one another. Please see the Authorship section of our editorial policies page below for more information: <https://www.nature.com/nature-research/editorial-policies/authorship#author-contribution-statements>

Response: Done